# Morphological and Molecular Analysis Identified a Subspecies of *Crassostrea ariakensis* (Fujita, 1913) along the Coast of Asia

**DOI:** 10.3390/genes15050644

**Published:** 2024-05-19

**Authors:** Ya Chen, Cui Li, Ruijing Lu, Haiyan Wang

**Affiliations:** 1Department of Marine Organism Taxonomy & Phylogeny, Institute of Oceanology, Chinese Academy of Sciences, Qingdao 266071, China; chenya@qdio.ac.cn (Y.C.); licui@qdio.ac.cn (C.L.); luruijing325@163.com (R.L.); 2University of Chinese Academy of Sciences, Beijing 101400, China; 3Chinese Academy of Sciences (CAS) Key Laboratory of Marine Ecology and Environmental Sciences, Institute of Oceanology, Chinese Academy of Sciences, Qingdao 266071, China; 4College of Life Sciences, Qingdao Agricultural University, Qingdao 266109, China

**Keywords:** *Crassostrea ariakensis*, *COI*, *16S rRNA*, subspecies, phylogenetic analysis, species delimitation

## Abstract

*Crassostrea ariakensis* (Fujita, 1913) is one of the most important economic and ecological oysters that is naturally distributed along the coast of Asia, separated by the Yangtze River estuary. They are usually compared as different populations, while there is no consensus on whether *C. ariakensis* in northern and southern areas should be considered as two species or subspecies. Here, we analyzed morphological characteristics, *COI*, *16s rRNA*, mitogenome sequences, and species delimitation analysis (ASAP and PTP) to resolve the intraspecific taxonomic status of the *C. ariakensis*. Phylogenetic and ASAP analysis highlight that *C. ariakensis* was divided into N-type and S-type. PTP was unable to differentiate between the two types of *C. ariakensis*. The divergence time of N-type and S-type *C. ariakinsis* is estimated to be 1.6 Mya, using the relaxed uncorrelated lognormal clock method. Additionally, significant morphological differences exist between the two groups in terms of the adductor muscle scar color. Despite these differences, the *COI* (0.6%) and *16S rRNA* (0.6%) genetic distance differences between N-type and S-type *C. ariakensis* has not yet reached the interspecific level. These results suggest that N-type and S-type *C. ariakensis* should be treated as different subspecies and renamed as *C. ariakensis ariakensis* subsp. nov and *C. ariakensis meridioyangtzensis* subsp. nov.

## 1. Introduction

Oysters present a challenge in classification because of the high level of phenotypic plasticity of the shell morphology [1,2]. *C. ariakensis* (Fujita, 1913), also known as the Jinjiang or Suminoe oyster, is one of the most economic and ecological oysters that is mainly naturally distributed in lower-salinity (10–25 ppt) estuaries in China, Japan, and Korea [3,4]. 

Previous molecular analyses have revealed two distinct clades of *C. ariakensis*, separated by the Yangtze River estuary, highlighting the unique genetic characteristics within this species and speculating the occurrence of reproduction isolation between these two divergent populations [4,5,6]. Kim et al. (2014) identified two divergent clades within *C. ariakensis* (the Fujian site-clade containing the southern population and the remaining sites-clade containing the northern population) using concatenated data from five mtDNA fragments (*16S rRNA*, *COI*, *COII*, *COIII*, and *Cytb*) [6]. Mitogenome studies have shown clear divergence among individuals within *C. ariakensis* into N-type and S-type groups, which is less apparent in other *Crassostrea* Sacco, 1897 species [7]. The whole genome data and resequencing analyses indicated that *C. ariakensis* along the Chinese coast has differentiated into southern and northern populations, and the degree of differentiation between the two distinct clades is similar to that of *C. gigas gigas* (Thunberg, 1793) and *C. gigas angulata* (Lamarck, 1819) [8,9]. Wang et al. (2004) proposed that this differentiation may be related to the existence of the Yangtze River estuary, acting as a natural barrier [5]. Kim (2014) suggested that the differentiation between northern and southern *C. ariakensis* can be attributed to various factors such as the Yangtze River’s freshwater influence, sea level changes in the paleo-East China and Yellow Seas, and biogeographical isolation [6].

Although the northern and southern *C. ariakensis* have traditionally been treated as distinct populations, there are still some uncertainties surrounding these two ecotypes. There are some morphological differences between the *C. ariakensis* from Northern China and *C. ariakensis* from Southern China. The adductor muscle scars of the northern *C. ariakensis* population are white, but are purple or brown in the southern *C. ariakensis* [5]. And the umbo cavities of the southern population from China is deeper than that of the northern population [5]. The conventional threshold for differentiating *Crassostrea* species typically involves a *COI* divergence greater than 2% [1,10]. Reciprocal hybridization experiments and intrapopulation crosses have been conducted to clarify the taxonomic status of northern and southern *C. ariakensis* as the same species [11], revealing incomplete local adaptation between the two environments [8]. These studies have revealed genetic differentiation between the two groups, suggesting limited gene flow and potential reproductive isolation. 

Existing studies generally treat the southern and northern groups of *C. ariakensis* as populations, and the taxonomy and population genetic analysis of *C. ariakensis* have predominantly focused on Chinese, Korean, and Japanese populations. Additionally, *C. ariakensis* from Southeast Asia are poorly known. Therefore, we analyzed morphological differentiation characteristics, Assemble Species by Automatic Partitioning (ASAP) [12], Poisson Tree Processes (PTP) [13], *COI*, *16S rRNA*, and mitochondrial genome sequences from all oysters collected from China, Korea, Japan, and Vietnam to provide further insights into whether *C. ariakensis* populations partitioned by the Yangtze River estuary should be considered as two species or subspecies. This study will help inform the selection and management of *C. ariakensis* and has a significant reference value for future oyster transplantation and the protection and restoration of oyster reefs.

## 2. Materials and Methods

### 2.1. Samples and Data Collection

A total of 354 *C. ariakensis* individuals were collected from 28 locations along the coast of China, Korean, Japan, and Vietnam (Figure 1 and Table 1). All samples were preserved in 95% ethanol immediately after collection and for subsequent analysis.

### 2.2. DNA Extraction, PCR Amplification, and Sequencing

Total genomic DNA was extracted from adductor muscle using the TIANamp marine animal DNA kit (Tiangen Biology, Beijing, China), following the manufacturer’s protocol. A fragment of *COI* was amplified with universal primers, LCO1490 and HCO2198 [14]. Primers of 16sar and 16sbr [15] were used to amplify a segment of the mitochondrial *16S rRNA* gene. 

The PCR amplification was performed in a 25 μL mixture under the following conditions: initial denaturation at 94 °C for 5 min, 30 cycles of 94 °C denaturation for 30 s, annealing at 48–51 °C for 1 min, extension at 72 °C for 1 min, and a final extension at 72 °C for 10 min. PCR products were verified on 1.5% agarose gels containing 0.2 μg/mL ethidium bromide, visualized under a UV transilluminator, and purified using DP214 Universal DNA Product Purification (Tiagen Biotech). The purified PCR products for mtDNA *COI* and *16S rRNA* were used as template for direct sequencing on an ABI Prism 3730 (Applied Biosystems, Waltham, MA, USA) automatic sequencer. Sequences were submitted to NCBI (http://www.ncbi.nlm.nih.gov/, accessed on 2 April 2024) under gene accession numbers PP575678-PP575743 for *COI*, and PP575655-PP575670 for *16S rRNA*.

### 2.3. Phylogenetic Analysis

The *COI*, *16S rRNA*, and mitogenome sequences obtained in this study and those of other Ostreidae Rafinesque, 1815 species from GenBank were subjected to phylogenetic analysis (Table 2). Initial multiple sequence alignments were performed using MAFFT 7 [16]. The single-gene (*COI* or *16S rRNA*) sequences were trimmed to the same length after alignment. DnaSP 6.0 [17] was utilized to estimate the total number of haplotypes (h) and their distribution in each location. Protein-coding genes’ (PCGs’) sequences were aligned in codon mode, using the invertebrate genetic code. The ribosomal RNAs’ (rRNAs’) sequences were aligned in normal mode. The conserved regions within the sequences were extracted using Gblocks [18] and concatenated to form a super-matrix based on the complete mitochondrial genomes. ModelFinder2 [19] was used to select the partition models under the Akaike’s information criterion. The Merge and Edge-linked modes were chosen.

Phylogenetic analyses were conducted using both maximum likelihood (ML) and Bayesian inference (BI) methods on the single-gene and super-matrix. *C. virginica* (Gmelin, 1791) or *Ostrea edulis* Linnaeus, 1758 was used as an outgroup. The best fitting models HKY + G were selected for COI and *16S rRNA* using ModelFinder [19] under Akaike’s information criterion. The ML analysis was performed in IQ-TREE [20] with 1000 ultrafast bootstrap replicates to infer the bootstrap values (BS) at each node. The Bayesian analysis was carried out in MrBayes v.3.2.6 [21] or BEAST v.1.10.4 [22]. Markov chain Monte Carlo (MCMC) searches were doubly run, with three independent runs being carried out for 30 million generations with a sampling frequency of 1000. Convergence was assessed by monitoring average standard deviations of split frequencies between three simultaneous runs (<0.01) and potential scale reduction factor (PSRF, close to 1.0). The program Tracer v1.7 [23] was applied to check all parameters for effective sampling size and unimodal posterior distribution. The first 25% of sampled trees were discarded as burn-in and the posterior probabilities were calculated from the remaining trees. FigTree v1.4.4, iTOL [24], and Adobe Illustrator were used to visualize and refine the phylogenetic trees. Pairwise sequence divergence among haplotypes and reference species was calculated using MEGA v11 [25], according to Kimura’s 2-parameter model.

### 2.4. Divergence Time Estimation

Based on 12PCG, BEAST v.1.10.4 [22] was used to estimate the species differentiation time, utilizing the relaxed uncorrelated lognormal clock method. Two calibration points were set to calibrate the divergence time of other nodes on the phylogenetic tree. Two reference divergence time points were retrieved from the database (http://fossilworks.org/bridge.pl, accessed on 8 March 2024) with 145.5 million years ago (Mya) being the time to the most recent common ancestor of *Crassostrea*, and 542 Mya Ma for TMRCA of Gastropoda and Bivalva [26,27]. The divergence time estimation using bivalve and Gastropod species is shown in Table 3, with *Katharina tunicata* (W. Wood, 1815) (NC_001636) of the Polyplacphora as the outgroup. The running parameter settings were as follows: The running algebra is 10^8^ generations, with sampling every 10^4^ generations. The model was set to GTR+G and the first 25% of data were discarded as burn-in. Tracer v.1.7 [23] was used to visualize and assess the effective population size of each parameter. TreeAnnotator v.2.6.2 [28] was used to estimate the 95% confidence interval for divergence time and they were identified in FigTree v.1.4.3.

### 2.5. Molecular Species Delimitation Analysis

Species delimitation of *C. ariakensis*, based on 12 PCG, was conducted using Assemble Species by Automatic Partitioning (ASAP) and the Poisson Tree Processes (PTPs). ASAP analysis was performed on the webserver (https://bioinfo.mnhn.fr/abi/public/asap/, accessed on 1 May 2024), based on Kimura’s 2-parameter model. The remaining parameters are set by the system default. The PTP model, a tree-based method, was employed to infer putative species boundaries on a given phylogenetic input tree. [13]. Initially, BEAST v.1.10.4 [22] was used to obtain the phylogenetic tree, with the best nucleotide substitution model being selected using jModelTest2.1 [29]. The BEAUti parameters were as follows: Yule model, relaxed uncorrelated lognormal clock, and 30,000,000 iterations for MCMC analysis, with sampling every 1000 steps. Tracer v1.7 was used to ensure the effective sample size (ESS > 200) of each parameter [23]. The maximum clade credibility tree was produced in TreeAnnotator v.2.6.2 [28]. PTP analysis was conducted on the webserver (https://species.h-its.org/ptp/, accessed on 1 May 2024) with the MCMC generations set to 1,000,000. Higher Bayesian support (BS) values on a node indicate that all descendants from this node are more likely to be from one species. The partition predicted using ASAP and PTP were selected for comparison with other molecular and morphological results.

## 3. Results

### 3.1. Shell Morphology

The morphology of *C. ariakensis* shells exhibited significant variation depending on environmental factors (Figure 2). There are some morphological differences between the southern *C. ariakensis* and northern *C. ariakensis*. The southern *C. ariakensis* displayed purple adductor muscle scars, whereas the northern populations exhibited white scars. 

### 3.2. COI Sequences

A 561bp *COI* sequence was sequenced for 342 oysters, generating a total of 66 haplotypes (Table A1). Hap1 is the common haplotype from the northern population. Hap11 is the common haplotype from the southern population. Oysters collected from Fenghua exhibited two haplotypes (Hap1 and Hap57), with Hap57 being shared with individuals from Shenzhen and Beihai (Table A1). The phylogenetic tree clearly separated populations geographically into the clade containing southern populations (Xiamen, Shantou, Shenzhen, Zhuhai, Huangmaotian, Chuandaozhen, Yangjiang, Zhanjiang, Baihai, Qinzhou, Fangchenggang, Hong Kong, and Vietnam) and another clade containing northern populations (Korea, Japan, Yingkou, Binzhou, Dongying, Guangrao, Weifang, Nantong, Shanghai, Haiyan, and Fenghua) (BP = 1) (Figure 3). 

The average genetic distance based on the mitochondrial *COI* gene sequences (using the Kimura 2-parameter model) between S-type *C. ariakensis* and N-type *C. ariakensis* is about 0.6%, which is lower than that observed between closely related sister species. Sequence divergence between *C. gigas angulata* and *C. gigas gigas* is 2.6%, divergence between *C. gigas gigas* and *C. sikamea* is 11.4%, and divergence between *C. hongkongensis* and *C. ariakensis* is 14.8–15.5% (Table 4). This level of divergence is similar to that observed within other *Crassostrea* species (0.74–1.48% in *C. gigas angulata* and 0.18–0.92% in *C. gigas gigas* [30] (Table 4)). Despite the clear division observed in the *COI* phylogenetic analysis between the southern and northern groups, the genetic distance indicates that these groups have not yet diverged to the extent of representing two separate species. These results indicate that *C. ariakensis* differentiation remains within the intraspecies level.

### 3.3. 16S rRNA Sequences

A 453 bp segment of *16S rRNA* was sequenced for 112 oysters, generating a total of 16 haplotypes (Table A2). Phylogenetic analysis was conducted using all *16S rRNA* haplotypes obtained in this study and other sequences from GenBank. *O. edulis* (AF052068) and *C. virginica* (AF092285) were used as an outgroup. They also constituted a monophyletic group (Figure 4 and Figure 5). In addition, the phylogenetic tree clearly separated populations geographically into the clade containing the southern population (Hap4,11,12,15; Xiamen, Shenzhen, Huangmaotian, Chuandaozhen, Yangjiang, Zhanjiang, Baihai, Qinzhou, Qukou, and Vietnam) and the clade containing the northern population (Hap1-3,5-10,13-15; Korea, Japan, Yingkou, Binzhou, Dongying, Weifang, Nantong, Shanghai, Haiyan, and Fenghua) (BS = 51; BP = 0.82).

The average genetic distance based on the mitochondrial *16S rRNA* gene sequence (using the Kimura 2-parameter model) between N-type *C. ariakensis* and S-type *C. ariakensis* is about 0.6%, which is higher than that observed within other *Crassostrea* species (0.26% in *C. gigas angulata*, 0.26% in *C. gigas gigas*, and 0.51% in *C. virginica* [30]) (Table 5). But it is lower than that observed between closely related sister species. Sequence divergence between *C. gigas angulata* and *C. gigas gigas* is 0.8%, divergence between *C. gigas gigas and C. sikamea* is 2.2%, and divergence between *C. hongkongensis and C. ariakensis* is 3.8% (Table 5). These genetic differences indicate that *C. ariakensis* should be two independent subspecies.

### 3.4. Mitogenome Sequences and Divergence Time Estimation

Phylogenetic analysis and divergence time estimation were conducted using mitochondrial genomic nucleotide sequences of 12 PCGs (except *atp8*) from all 46 individuals listed in Table 3. The topological structures of Bayesian trees and maximum likelihood trees constructed based on mitogenome are basically consistent. The relationship among individuals within *C. ariakensis* are clearly diverged into N-type and S-type, as the *COI* and *16S rRNA* has also demonstrated. The phylogenetic relationship of the mitogenome tree is clearer and has higher Bayesian posterior probabilities (BP = 1) and maximum likelihood bootstrap support (BS = 100) values than the single-gene tree (Figure 6). The divergence time of the TMRCA for the N-type and S-type *C. ariakensis* to be 1.6 Mya with a 95% confidence interval of 0.92–2.60 Mya (Figure 7).

### 3.5. Species Delimitation Analysis

In the species delimitation analysis of ASAP (Figure 8A), 13 distinct species subsets were delineated with a score of 7.0, which is consistent with the result of the phylogenetic analysis. *C. gigas angulata* and *C. gigas gigas* are clustered in the different subsets. *C. ariakensis* are also clearly diverged into N-type and S-type. The PTP analysis results suggest that *C. gigas angulata* (BS = 1.00) and *C. gigas gigas* (BS = 1.00) belong to distinct subsets, whereas the *C. ariakensis* from both the northern and southern regions are recognized as the same subset (BS = 0.956, Figure 8B). Both species delimitation method analyses indicate that the *C. ariakensis* belongs to the level of intraspecific differentiation.

## 4. Discussion

### 4.1. Identification of C. ariakensis Subspecies

Systematics

Phylum Mollusca

Class Bivalvia

Order Ostreida Férussac, 1822

Family Ostreidae Rafinesque, 1815

Subfamily Crassostreinae Scarlato and Starobogatov, 1979 

Genus *Crassostrea* Sacco, 1897

Species *C. ariakensis* (Fujita, 1913)

Subspecies *C. ariakensis ariakensis* subsp. nov.

*C. ariakensis meridioyangtzensis* subsp. nov.

The shell of *C. ariakensis* is variable in shape, often appearing elongated ovate or slightly ovoid. The left valve is thicker and more convex than the right valve. The ventral margin is rounded. The surface is covered by platy growth lamellae without any strong plications. The outer valves are yellow-brown or gray. The internal valves are white. Adductor muscle scars are kidney shaped, close to the posterior valve margin, and closer to the ventral margin than to the hinge. *C. ariakensis ariakensis* subsp. nov. can be readily distinguished from *C. ariakensis meridioyangtzensis* subsp. nov. by having white adductor muscle scars. The adductor muscle scars of the *C. ariakensis meridioyangtzensis* subsp. nov. are purple or brown. It is possible that the long-term geographical isolation of two populations has led to their morphological changes and that they may be two subspecies [31]. 

### 4.2. Distribution of C. ariakensis Subspecies

In this study, we collected and sequenced a large number (n = 354) of oysters from 28 sites. Previous studies of the taxonomy and population genetic analysis of *C. ariakensis* have focused mainly on Chinese, Korean, and Japanese populations. In this study, *C. ariakensis* has also been observed in Vietnam and is limited to the low-salinity estuarine.

Our results indicate that all oysters from the 14 northern sites (Seomjin River, Sacheon Kawha River, Kangwha-do, Ariake Bay, Itoki River, Yingkou, Binzhou, Dongying, Guangrao, Weifang, Nantong, Shanghai, Haiyan, and Fenghua) are *C. ariakensis ariakensis* (N-type *C. ariakensis*) and that all oysters from the 14 southern sites (Xiamen, Shantou, Shenzhen, Zhuhai, Huangmaotian, Chuandaozhen, Yangjiang, Zhanjiang, Baihai, Qinzhou, Fangchenggang, Hong Kong, Hainan, and Vietnam) are *C. ariakensis meridioyangtzensis* subsp. nov. (S-type *C. ariakensis*). The Yangtze River may be responsible for the distribution and genetic differences of *C. ariakensis ariakensis* subsp. nov. and *C. ariakensis meridioyangtzensis* subsp. nov. As is well known, the Yangtze River estuary is a barrier for the distribution of many marine invertebrates [32], possibly because it hinders the dispersal of larvae and is a junction of cold and warm temperatures in the north and south. Nevertheless, extensive sampling, especially in and around the Yangtze River and Southeast Asia, may help to better define the distribution bordering of *C. ariakensis ariakensis* subsp. nov. and *C. ariakensis meridioyangtzensis* subsp. nov. It exhibits gregarious behavior, attaching to substrates primarily with the left valve. *C. ariakensis meridioyangtzensis* subsp. nov. are often found cohabiting with *C. hongkongensis*. 

### 4.3. Relationship between C. ariakensis ariakensis subsp. nov. and C. ariakensis meridioyangtzensis subsp. nov.

In this study, we analyzed *COI*, *16s rRNA*, and mitochondrial genome to determine whether the *C. ariakensis* in the north and south should be considered as two species or subspecies. Phylogenetic analysis can clearly reveal that the *C. ariakensis* from 28 locations were clustered into two typical groups (northern and southern), which was consistently correlated to their geographical distribution. Our finding was consistent with those of previous molecular taxonomic studies which found that *C. ariakensis* from Northern China, Korea, and Japan were more closed related [4,6,33]. The *C. ariakensis* in the south of China is closer to the *C. ariakensis* oyster in Vietnam. Several populations near the Yangtze River estuary (Fenghua, Haiyan, Shanghai, and Nantong) have close genetic relationships with several populations in the north. This may be due to the fact that the freshwater influx from the Yangtze River extends towards Jizhou Island in summer [32], facilitating larval dispersal, and enhancing connectivity with northern populations under the influence of the warm current in the Yellow Sea. However, the complex ocean current environment, including opposing coastal currents and the Taiwan Warm Current south of the Yangtze River estuary, impedes communication with the southern group, reinforcing the genetic differentiation between northern and southern populations. The Fenghua population contains haplotypes of two lineages, with one individual corresponding to a haplotype shared with the southern group. It is speculated that the Fenghua is the location of secondary contact between the northern and southern lineages, which is akin to the secondary contact zone observed in other species distributed in this region [34]. 

The phylogenetic analysis indicated that the phylogenetic tree constructed from the mitochondrial genome exhibited higher support values compared to analyses based on single-gene markers. It is generally believed that multiple gene segments contain richer genetic information, thus providing a more robust depiction of the evolutionary status of a species in phylogenetic analysis. Different gene markers may yield varying results. Therefore, obtaining more comprehensive genetic information from the genome and transcriptome levels for *C. ariakensis* is crucial to accurately reflect its classification status. The results of species delimitation methods were consistent with those of morphological and phylogenetic identification. The method of comprehensive identification of species based on morphological characteristics, phylogenetic topology, and molecular species delimitation technology has significantly improved the efficiency and accuracy of species identification.

The genetic distance differences between N-type and S-type *C. ariakensis* have not yet reached the interspecific level observed within the *Crassostrea* genus. The isolation of these southern and northern populations reflects habitat specificity, mirroring the patterns observed in *C. gigas angulata* and *C. gigas gigas* [30]. The average genetic distance calculated from the mitochondrial *16S rRNA* gene sequence (using the Kimura 2-parameter model) between N-type and S-type *C. ariakensis* is 0.6%, which is similar to that observed between closely related sister species (0.8%, *C. gigas angulata* and *C. gigas gigas*). The average genetic distance between N-type and S-type *C. ariakensis* is 0.6% in *COI*, which is lower than that observed between *C. gigas angulata* and *C. gigas gigas* (2.6% in *COI*). The level of genetic distance being less than 1% has hindered N-type and S-type *C. ariakensis* to be considered as two distinct species [31].

This study estimates that N-type and S-type *C. ariakensis* began to diverge approximately 0.92–2.60 Mya, based on fossil calibration. Li et al. (2021) estimated a more recent divergence time between the southern and northern *C. ariakensis* populations, ranging from 0.14 to 0.63 Mya [8]. Additionally, the Pairwise Sequentially Markovian Coalescent analysis showed that the effective population sizes of the northern and southern *C. ariakensis* groups began to separate around 0.1 Mya [9]. This suggests that the two ecotypes have been evolving independently for a substantial period, potentially leading to local adaptation and genetic differentiation between N-type and S-type *C. ariakensis*. Multiple lines of evidence, including fossil records, mitochondrial DNA, and whole-genome analyses, converge on a divergence time estimate ranging from 0.1 to 2.6 Mya for N-type and S-type *C. ariakensis* [8,9]. This indicates that these two ecotypes have been evolving in isolation for an extended period, which may have important implications for their ecology, physiology, and potential for their successful introduction outside of their native ranges.

Based on the stable genetic differentiation and geographical isolation observed in the northern and southern populations of *C. ariakensis*, we suggest that N-type and S-type *C. ariakensis* should be recognized as distinct subspecies and should be renamed as *C. ariakensis ariakensis* subsp. nov. *and C. ariakensis meridioyangtzensis* subsp. nov. However, the extent of divergence and the underlying mechanisms driving the apparent ecotypic differences remain unclear. It is possible that environmental factors, such as temperature regimes and food availability, play key roles in shaping the local adaptation and performance of these oyster populations. Further research is needed to fully elucidate the evolutionary and ecological relationships between the northern and southern *C. ariakensis* ecotypes. Understanding these will provide valuable insights into the adaptive strategies of this species and inform effective conservation and management strategies in response to environmental changes and anthropogenic impacts.

## 5. Conclusions

Research has shown distinct genetic variation in *C. ariakensis* between northern and southern populations. We analyzed morphological differentiation characteristics, *COI*, *16S rRNA*, mitochondrial genome sequences, and species delimitation analysis to resolve the taxonomic status of the *C. ariakensis* populations separated by the Yangtze River estuary. The results highlighted that the populations of *C. ariakensis* were divided into N-type and S-type clades. The northern Chinese populations were found to be more closely related to the populations in Korea and Japan, whereas the southern Chinese populations exhibited a closer relationship with the populations in Vietnam. Additionally, significant morphological differences exist between the two groups, particularly in terms of the adductor muscle scar color. Despite these differences, the genetic distance differences between N-type and S-type *C. ariakensis* have not yet reached the interspecific level observed within the *Crassostrea* genus. Consequently, we suggest that N-type and S-type *C. ariakensis* should be treated as different subspecies and be renamed as *C. ariakensis ariakensis* subsp. nov. and *C. ariakensis meridioyangtzensis* subsp. nov.

## Figures and Tables

**Figure 1 genes-15-00644-f001:**
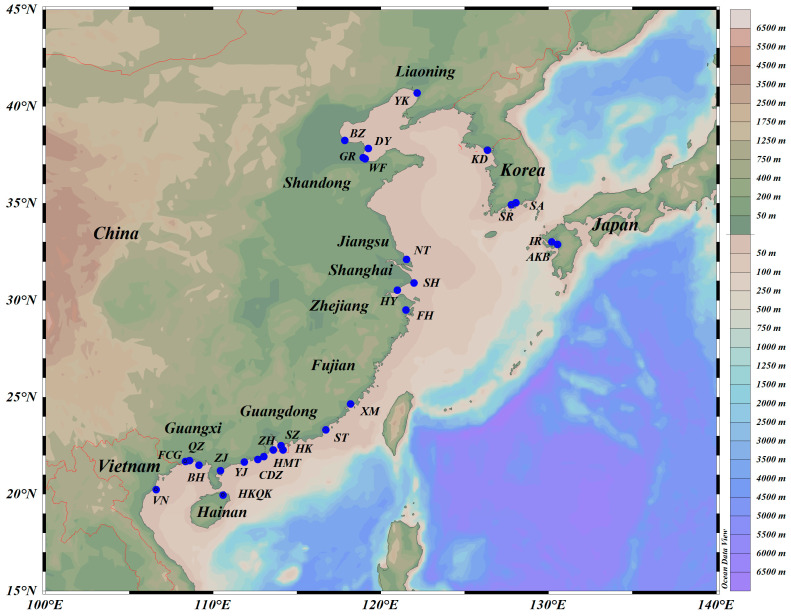
Map of sampling sites for 28 populations of *C. ariakensis*.

**Figure 2 genes-15-00644-f002:**
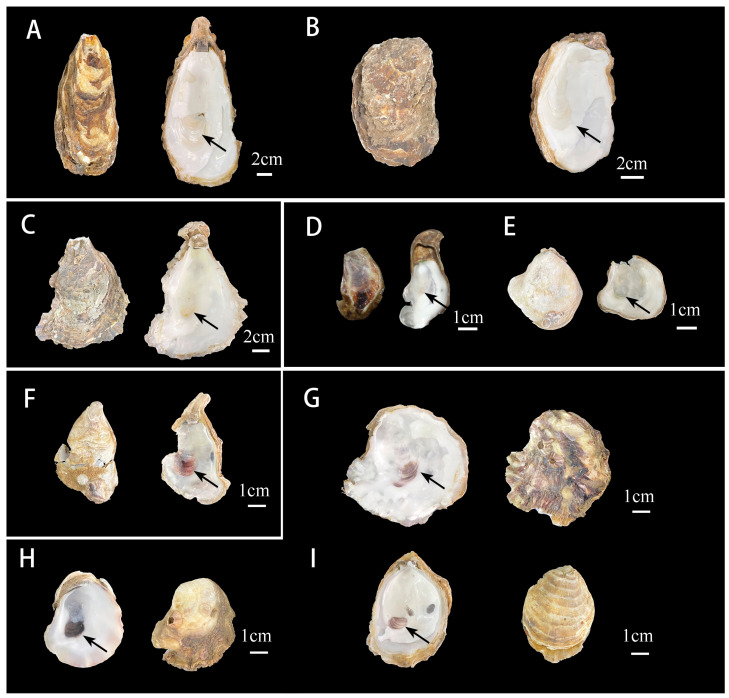
Shell morphology of the representative *C. ariakensis* in this study. (**A**,**B**) *C. ariakensis* from Binzhou; (**C**) *C. ariakensis* from Weifang; (**D**) *C. ariakensis* from Haiyan; (**E**) *C. ariakensis* from Shanghai; (**F**) *C. ariakensis* from Taishan, MBM287905; and (**G**–**I**) *C. ariakensis* from Yangjiang, MBM287906-08.

**Figure 3 genes-15-00644-f003:**
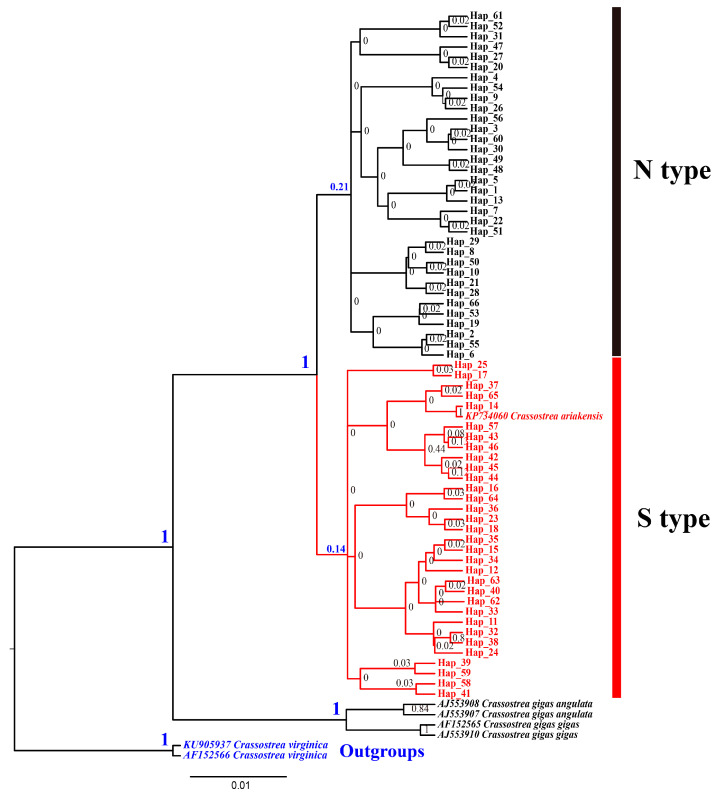
The phylogenetic tree based on *COI* gene. Numbers near the nodes are reporting the bootstrap values of the Bayesian phylogenetic analyses.

**Figure 4 genes-15-00644-f004:**
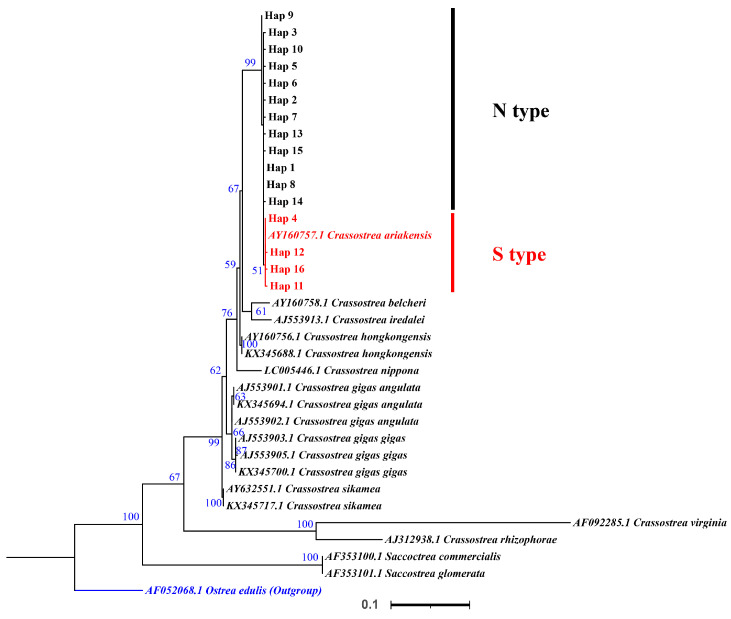
IQtree of haplotypes based on *16S rRNA* gene. Numbers near the nodes are reporting the bootstrap values of the maximum likelihood phylogenetic analyses. The branch number is bootstrap value > 50%.

**Figure 5 genes-15-00644-f005:**
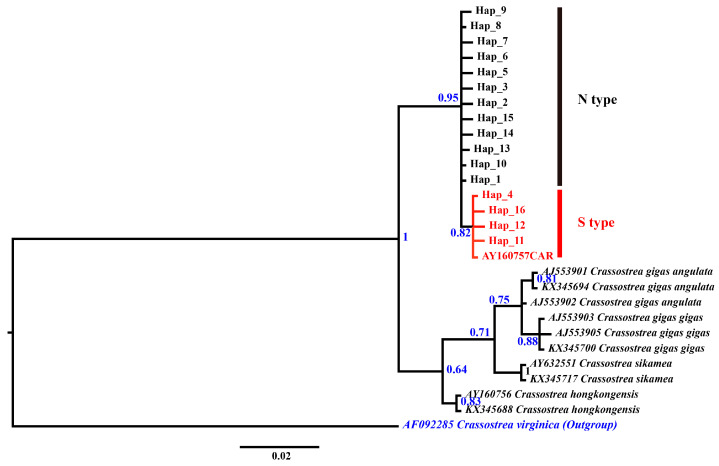
Bayesian inference (BI) tree of haplotypes based on *16S rRNA* gene. The numbers near the internal branches represent percent bootstrap support values based on the posterior probability with the Bayesian method.

**Figure 6 genes-15-00644-f006:**
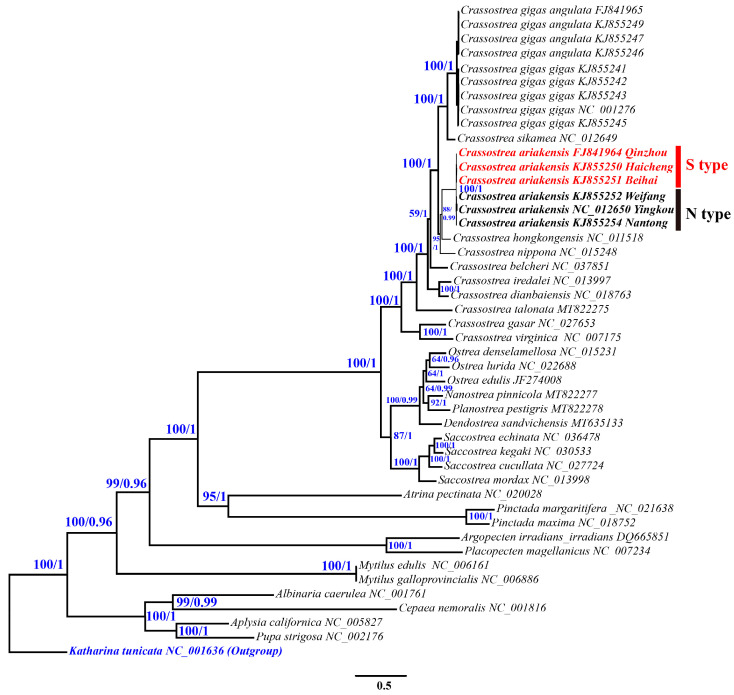
Phylogenetic reconstruction from maximum likelihood and Bayesian analyses of mitogenome sequences. Numbers near the nodes are branch support values (Bayesian posterior probabilities followed by maximum likelihood bootstrap support values).

**Figure 7 genes-15-00644-f007:**
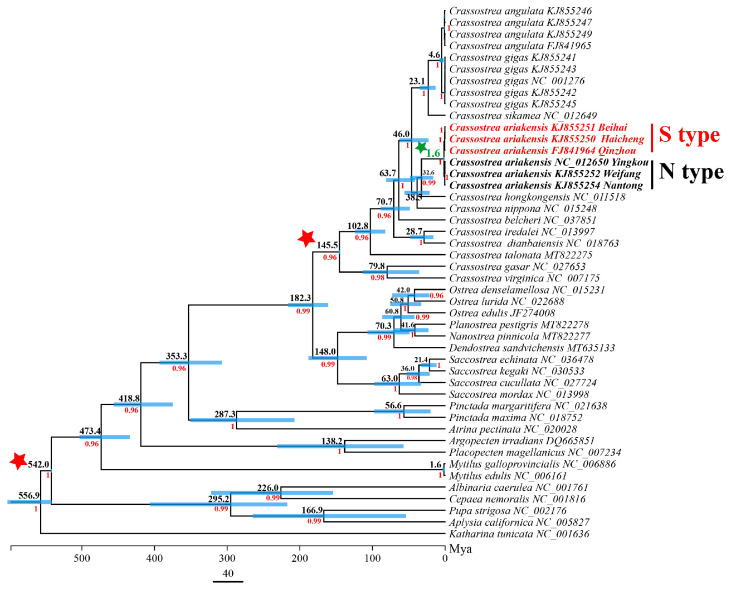
Estimates of divergence times based on 12 protein-coding genes. Numbers near the nodes indicate the median ages and blue bars indicate 95% highest posterior density intervals. Calibration points are marked using a red pentagram. The green pentagram represents the estimated divergence time of *C. ariakensis*.

**Figure 8 genes-15-00644-f008:**
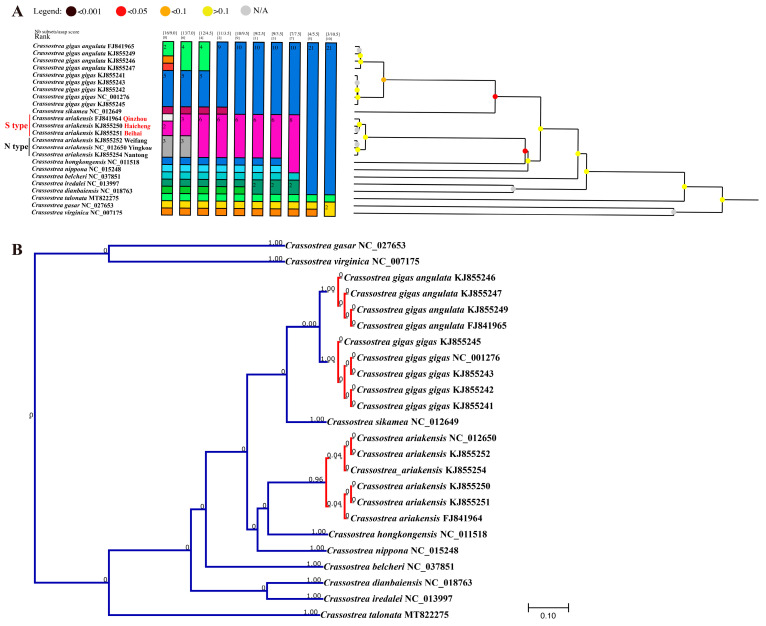
Species delimitation analysis of *C. ariakensis*, based on mitochondrial genomic nucleotide sequences of 12 PCGs (except *atp8*). (**A**) ASAP analysis results. The different color rectangles indicate different species. (**B**) PTP species delimitation results. Higher Bayesian support values on a node indicate that all descendants from this node are more likely to be from one species. The red lines clustered in the tree with higher support values represent the same species.

**Table 1 genes-15-00644-t001:** Locations and numbers of *C. ariakensis* collected and sequenced in this study.

Location	*n*	Longitude [Degrees East]	Latitude [Degrees North]
SR: Seomjin River, South Korea *	10	121.76	34.92
SA: Sacheon Kawha River, South Korea *	5	128.04	35.02
KD: Kangwha-do, South Korea *	3	126.35	37.74
AKB: Ariake Bay, Japan *	35	130.52	32.88
IR: Itoki River, Japan *	3	130.18	33.02
YK: Yingkou, Liaonning, China *	2	122.15	40.69
BZ: Binzhou, Shandong, China	5	117.85	38.25
DY: Kenli, Dongying, Shandong, China	11	119.24	37.83
GR: Guangrao, Dongying, Shandong, China *	36	118.94	37.35
WF: Weifang, Shandong, China	6	119.05	37.29
NT: Nantong, Jiangsu, China	39	121.52	32.11
SH: Shanghai, China	8	121.97	30.89
HY: Haiyan, Zhejiang, China	17	120.98	30.53
FH: Fenghua, Zhejiang, China	5	121.49	29.50
XM: Xiamen, Fujian, China *	34	118.19	24.66
ST: Shantou, Guangdong, China	13	116.72	23.33
SZ: Shenzhen, Guangdong, China	14	114.04	22.50
ZH: Zhuhai, Guangdong, China *	11	113.58	22.29
HMT: Huangmaotian, Taishan, Guangdong, China	3	113.02	21.94
CDZ: Chuandaozhen, Taishan, Guangdong, China	3	112.65	21.80
YJ: Yangjiang, Guangdong, China	2	111.85	21.66
ZJ: Zhanjiang, Guangdong, China	39	110.43	21.21
BH: Beihai, Guangxi, China	14	109.16	21.51
QZ: Maoweihai, Qinzhou, Guangxi, China *	10	108.58	21.74
FCG: Fangchenggang, Guangxi, China *	19	108.34	21.69
HK: Pearl River Delta, Hong Kong, China *	3	114.16	22.29
HKQK: Qukou, Haikou, China	1	110.59	19.95
VT: Vietnam	3	106.60	20.24

* Locations with asterisk are from the references.

**Table 2 genes-15-00644-t002:** Information of specimens and sequences from GenBank analyzed in this study.

Species	GenBank Accession Number	
	*COI*	*16S rRNA*
*C.* *ariakensis*	FJ743512-27 KP734018-62	KX345399-410 FJ743503-07
	EU007496-98	LC005447
	EU007503-05	EU672835 * NC_012650 *
	EU672835 * NC_012650 *	AY632546-48 KJ855250-52, KJ855254
	KX345411-28	HQ660979-80 KC847118
	AY632559-66	FJ841964 *
	HQ661020-21	
	EU007493 FJ841964 * AY160752-54	
*C.* *ariakensis*	AF300617, KP734060	AY160757
*C. hongkongensis* (Lam and B. Morton, 2003)	AJ553912, KP976208	AY160756, KX345688
*C. gigas angulata*	AJ553908, AJ553907, KP216805	AJ553901, AJ553902, KX345694
*C. gigas gigas*	AF152565, AJ553910, KP099016	AJ553903, AJ553905, KX345700
*C. sikamea* (Amemiya, 1928)	AF152568, AB904878	AY632551, KX345717
*C.* *virginica*	AF152566, KU905937	AF092285
*C. rhizophorae* (Guilding, 1828)	KP455050	AJ312938
*C. belcheri* (G. B. Sowerby II, 1871)	AY160755	AY160758
*C. iredalei* (Faustino, 1932)	AY038078	AJ553913
*C. nippona* (Seki, 1934)	--	LC005446
*Saccostrea commercialis* (Iredale and Roughley, 1933)	--	AF353100
*O. edulis*	AF540599	AF052068
*S. cuccullata* (Born, 1778)	AY038076	--
*S. glomerata* (A. Gould, 1850)	--	AF353101

* Accession numbers with asterisk are from the mitochondrial complete sequences.

**Table 3 genes-15-00644-t003:** Mitochondrial genomic information of species used for phylogenetic analysis and divergence time estimation.

Taxon	Species	GenBank Accession Number
Bivalvia	*C. gigas angulata*	KJ855247
	*C. gigas angulata*	KJ855249
	*C. gigas angulata*	KJ855246
	*C. gigas angulata*	FJ841965
	*C. gigas gigas*	NC_001276
	*C. gigas gigas*	KJ855243
	*C. gigas gigas*	KJ855245
	*C. gigas gigas*	KJ855242
	*C. gigas gigas*	KJ855241
	*C. sikamea*	NC_012649
	*C.* *ariakensis*	NC012650
	*C.* *ariakensis*	KJ855252
	*C.* *ariakensis*	KJ855254
	*C.* *ariakensis*	KJ855250
	*C.* *ariakensis*	KJ855251
	*C.* *ariakensis*	FJ841964
	*C. hongkongensis*	NC_011518
	*C. nippona*	NC_015248
	*C. becheri*	NC_037851
	*C. iredalei*	NC_013997
	*C. dianbaiensis* J.-J. Xia, X.-Y. Wu, S. Xiao, and Z. Yu, 2014	NC_018763
	*C. talonata* X.-X. Li and Z.-Y. Qi, 1994	MT822275
	*C. gasar* (Dautzenberg, 1891)	NC_027653
	*C. virginica*	NC_007175
	*Nanostrea exigua pinnicola* (Pagenstecher, 1877)	MT822277
	*Planostrea pestigris* (Hanley, 1846)	MT822278
	*Dendostrea sandvichensis* (G. B. Sowerby II, 1871)	MT635133
	*O. denselamellosa* Lischke, 1869	NC_015231
	*O. edulis*	JF274008
	*O. lurida* P. P. Carpenter, 1864	NC_022688
	*S. echinata* (Quoy and Gaimard, 1835)	NC_036478
	*S. kegaki* Torigoe and Inaba, 1981	NC_030533
	*S. cucullata*	NC_027724
	*S. mordax* (Gould, 1850)	NC_013998
	*Pinctada maxima* (Jameson, 1901)	NC_018752
	*P. margaritifera* (Linnaeus, 1758)	NC_021638
	*Atrina pectinate* (Linnaeus, 1767)	NC_020028
	*Mytilus edulis* Linnaeus, 1758	NC_006161
	*M. galloprovincialis* Lamarck, 1819)	NC_006886
	*Argopecten irradians* (Lamarck, 1819)	DQ665851
	*Placopecten magellanicus* (Gmelin, 1791)	NC_007234
Gastropoda	*Albinaria caerulea* (Deshayes, 1835)	NC_001761
	*Aplysia californica* J. G. Cooper, 1863	NC_005827
	*Cepaea nemoralis* (Linnaeus, 1758)	NC_001816
	*Pupa strigosa* (A. Gould, 1859)	NC_002176
Polyplacphora	*K. tunicata*	NC_001636

**Table 4 genes-15-00644-t004:** Pairwise sequence divergence among *COI* haplotypes observed in this study.

Species	*Car_N*	*Car_S*	*Can*	*Cgi*	*Csi*	*Cvi*	*Cbe*	*Cir*	*Chk*	*Oed*	*Scu*	*Crh*
*Car_N*		0.002	0.021	0.021	0.021	0.029	0.023	0.022	0.021	0.029	0.034	0.030
*Car_S*	0.006		0.021	0.021	0.020	0.030	0.024	0.022	0.020	0.029	0.033	0.029
*Can*	0.164	0.161		0.007	0.016	0.025	0.023	0.022	0.019	0.031	0.033	0.027
*Cgi*	0.159	0.156	0.026		0.016	0.025	0.023	0.021	0.019	0.030	0.033	0.027
*Csi*	0.164	0.161	0.105	0.114		0.026	0.022	0.023	0.019	0.029	0.032	0.027
*Cvi*	0.280	0.283	0.235	0.238	0.239		0.028	0.028	0.029	0.030	0.034	0.023
*Cbe*	0.199	0.202	0.189	0.188	0.171	0.256		0.022	0.024	0.029	0.032	0.029
*Cir*	0.175	0.177	0.175	0.173	0.194	0.255	0.185		0.022	0.029	0.030	0.030
*Chk*	0.152	0.148	0.138	0.137	0.147	0.260	0.203	0.166		0.030	0.034	0.029
*Oed*	0.292	0.289	0.300	0.292	0.271	0.293	0.278	0.262	0.291		0.029	0.031
*Scu*	0.324	0.321	0.310	0.306	0.312	0.347	0.318	0.276	0.323	0.269		0.035
*Crh*	0.282	0.279	0.256	0.259	0.254	0.183	0.274	0.285	0.275	0.310	0.352	

*Car_N*: Northern *C. ariakensis*; *Car_S*: Southern *C. ariakensis*; *Car*: *C. ariakensis*; *Can*: *C. gigas angulata*; *Cgi*: *C. gigas gigas*; *Csi*: *C. sikamea*; *Cvi*: *C. virginica*; *Cbe*: *C. belcheri*; *Cir*: *C. iredalei*; *Chk*: *C. hongkongensis*; *Oed*: *O. edulis*; *Scu*: *S. cuccullata*; *Crh*: *C. rhizophorae*. Standard error estimate(s) are shown above the diagonal. Analyses were conducted using the Kimura 2-parameter model.

**Table 5 genes-15-00644-t005:** Pairwise sequence divergence among *16S rRNA* haplotypes observed in this study.

Species	*Car_N*	*Car_S*	*Can*	*Cgi*	*Csi*	*Cvi*	*Crh*	*Cbe*	*Cir*	*Chk*	*Sco*	*Oed*	*Sgl*	*Cni*
*Car_N*		0.003	0.012	0.012	0.011	0.030	0.024	0.013	0.012	0.010	0.025	0.022	0.025	0.014
*Car_S*	0.006		0.011	0.012	0.011	0.029	0.024	0.012	0.012	0.010	0.025	0.022	0.025	0.014
*Can*	0.053	0.051		0.004	0.007	0.029	0.023	0.011	0.011	0.007	0.026	0.023	0.026	0.011
*Cgi*	0.056	0.055	0.008		0.007	0.029	0.023	0.012	0.011	0.008	0.027	0.023	0.027	0.011
*Csi*	0.050	0.047	0.018	0.022		0.030	0.023	0.010	0.010	0.008	0.025	0.023	0.025	0.012
*Cvi*	0.272	0.270	0.264	0.262	0.272		0.025	0.030	0.032	0.030	0.039	0.035	0.039	0.030
*Crh*	0.190	0.187	0.173	0.173	0.176	0.209		0.024	0.023	0.024	0.032	0.029	0.032	0.026
*Cbe*	0.059	0.056	0.048	0.056	0.044	0.284	0.190		0.010	0.010	0.027	0.024	0.027	0.013
*Cir*	0.059	0.060	0.047	0.050	0.038	0.299	0.183	0.041		0.010	0.027	0.024	0.027	0.012
*Chk*	0.036	0.040	0.022	0.025	0.024	0.272	0.179	0.041	0.038		0.026	0.022	0.026	0.010
*Sco*	0.203	0.201	0.212	0.214	0.202	0.392	0.299	0.227	0.216	0.210		0.028	0.000	0.027
*Oed*	0.166	0.163	0.182	0.183	0.172	0.338	0.241	0.182	0.182	0.175	0.245		0.028	0.026
*Sgl*	0.203	0.201	0.212	0.214	0.202	0.392	0.299	0.227	0.216	0.210	0.000	0.245		0.027
*Cni*	0.072	0.076	0.046	0.045	0.050	0.276	0.203	0.067	0.064	0.041	0.228	0.203	0.228	

*Car_N*: Northern *C. ariakensis*; *Car_S*: Southern *C. ariakensis*; *Can*: *C. gigas angulata*; *Cgi*: *C. gigas gigas*; *Csi*: *C. sikamea*; *Cvi*: *C. virginica*; *Crh*: *C. rhizophorae*; *Cbe*: *C. belcheri*; *Cir*: *C. iredalei*; *Chk*: *C. hongkongensis*; *Sco*: *S. commercialis*; *Oed*: *O. edulis*; *Sgl*: *S. glomerata*; *Cni*: *C. nippona*. Standard error estimate(s) are shown above the diagonal. Analyses were conducted using the Kimura 2-parameter model.

## Data Availability

All sequences are available from GenBank under the accession numbers PP575655-PP575670 and PP575678-PP575743.

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
