# Peer review of "Morphological and Molecular Analysis Identified a Subspecies of Crassostrea ariakensis (Fujita, 1913) along the Coast of Asia"

_genes, 2024, doi:10.3390/genes15050644_

Round 1
Reviewer 1 Report
Comments and Suggestions for Authors
line 67-68:Quoted references do not go beyond population level. Your aim, so it seems, is to stabilise the status by deciding between population, subspecies (many definitions in circulation, be warned!) and species. Say so explicitly.
line 72-73:You cannot put the main conclusion in the Introduction!
line 214-215:Briefly explain what you consider to be subspecies, in your specific case. It cannot be solely defined on the basis of genetic distance. Important issue is a reproductive isolation (most frequently, geographic). Any hybrids?
line 251-252: Simplify if possible to yangtzensis
line 358-359: Simply C. a. ariakensis subsp.nov and C. a. yangtzensis subsp.nov.
Comments on the Quality of English LanguageRequires moderate editing.
Reviewer 2 Report
Comments and Suggestions for Authors
Dear authors,
1) There is enormous plasticity in the group, variation in the umbo is mentioned, but it is not presented and in the photos I see enormous plasticity in it.
2) The 2% differentiation rate is not achieved, the results indicate 0.6% which is much lower than expected.
3) Subspecies is a concept that is increasingly out of use because it is not logical. How many differences are necessary for one species to be different from another? Isn't one enough? Why then, if there is a difference, not consider a new species?
4) Analyzing your figures 3, 4 and 5 I see that the supports are very low. You are basing your results on topology and are not seeing that the support presented does not prove anything that was said. Taking this into consideration, figures 6 and 7 are put in check. How did you get results with such high support if in individual analyzes the value is so low? Why did the number of specimens tested in the analysis decrease? Look at the genetic distance between the brances, which is practically 0 in the clade. I can't believe these support values ​​given everything that was presented before.
5) In these types of cases, the ideal would be to carry out species delimitation. There are several methods, ABDG, GMYC, BPTP, among others. These methods are much more appropriate to solve the case they are trying to solve.
6) Analyzes using time with genes that evolve so quickly are not appropriate.
7) Never forget to include author and year for the first time a species or genus is mentioned in the text. In addition to placing the zoobank code for nomenclatural acts.
With all this in mind, I believe that the objective of the manuscript must be rethought and it must undergo major modifications to obtain better results that prove what the authors hope to find.
Round 2
Reviewer 2 Report
Comments and Suggestions for Authors
Dear authors,
There was a great improvement in the manuscript in several aspects. However, I still see some details that deserve attention.
- It is GMYC not GWYC.
- These analyzes (GMYC and ASPA) must be performed with only samples of the presumed species to be evaluated and the most related species (C. gigas; C. angulata; C. hongkongensis; C. niponica; and C. sikamea). Selection of highly distant outgroups cause erosion on the resolving power of the analyses, since they go against the theoretical assumptions of the original methodology, use DAMBE to select the unique haplotypes for the analysis. Therefore, the analyzes must be redone.
- I still do not agree with the description of the subspecies, I feel that the description of two subspecies is being forced, while for me it is only populational variation, as other authors also think. As far as I can see, there is still gene flow between these populations given the low genetic distance.On the other hand, there is some morphological differentiation, so I don't see why not describe one of the populations as a new species. Speciation may be recent and may not be detected by molecular methods. I think the authors could discuss this possibility and think about describing a species.The diagnosis is ready, just give the name. I believe this would be a breakthrough for practical applications and scientific research value.
But I leave that decision up to the author and editor.
I invite the author to read the following paper about subspecies:
https://onlinelibrary.wiley.com/doi/full/10.1002/ece3.9069
